# Which companies dominate the packaged food supply of New Zealand and how healthy are their products?

Sally Mackay[1]*, Helen Eyles[1,2], Teresa Gontijo de Castro[1], Leanne Young[2], Cliona Ni Mhurchu[2], Boyd Swinburn[1]

1 Department of Epidemiology and Biostatistics, School of Population Health, The University of Auckland, Auckland, New Zealand, 2 National Institute for Health Innovation, The University of Auckland, Auckland, New Zealand

* sally.mackay@auckland.ac.nz

**Data Availability Statement:** Because of the commercial and legal restrictions to the use of copyrighted material it is not possible to share data openly which reveal the product or company

## Abstract

Improvement of national food supplies are an opportunity to improve a country's health. Our aim was to identify the major food companies manufacturing packaged foods and non-alcoholic beverages available in New Zealand supermarkets in 2018; to assess the healthiness of products using (1) the Health Star Rating (HSR) system, (2) Australian Dietary Guidelines classification (core/discretionary), and (3) by level of processing; to compare the healthiness of products displaying and not displaying the HSR and; to assess potential for food reformulation within selected food sub-categories. Information on packaged foods was obtained from the Nutritrack supermarket database. Companies that manufactured each food and brand were identified using company websites and the New Zealand companies register. In total, 13,506 packaged products were mapped to 1,767 brands and 1,214 companies. Based on market share of products available for sale (Euromonitor data), there were 22 dominating companies producing 31% of products and 17% of brands. Fifty-nine percent of products were classified as unhealthy (HSR <3.5/5 stars), 53% as discretionary, and 69% as ultra-processed. Products displaying the HSR on the package had a higher mean HSR ±SD than if the HSR was not displayed (3.2±1.3 *versus* 2.5±1.4, p = 0.000). Efforts to improve the healthiness of products should be directed to the 22 food companies dominating this market share, particularly in the core foods groups which are currently less likely to meet Heart Foundation reformulation targets (bread, breakfast cereals, cheese, canned baked beans, yoghurt). The New Zealand supermarket packaged food supply included in the Nutritrack database is dominated by a small number of companies and is mostly unhealthy. Government leadership is required to improve the healthiness of the packaged food supply and provide adequate information to consumers. This includes interventions setting reformulation targets for core food groups, setting population nutrient intake targets and mandating that the HSR is displayed on all products.

names but unredacted versions of the dataset are available with a licensed agreement that they will be restricted to non-commercial use. For access to Nutritrack, please contact the National Institute for Health Innovation at the University of Auckland at enquiries@nihi.auckland.ac.nz.

**Funding:** This research was supported by the National Health and Medical Research Council (NHMRC) funded Centre of Research Excellence in Reducing Salt Intake using Food Policy Interventions (APP1117300). The opinions, analysis and conclusions in this paper are those of the authors and should not be attributed to the NHMRC. The Nutritrack data collections and database are funded by a Health Research Council of New Zealand programme grant (18/672. The funders had no role in study design, data collection and analysis, decision to publish, or preparation of the manuscript.

**Competing interests:** The authors have declared that no competing interests exist.

## Introduction

The burden of disease attributable to dietary factors among adults aged 25 years or older is 11 million deaths and 255 million disability adjusted-life years (DALYs) [1, 2]. High intake of sodium and low intakes of whole grains and fruit accounted for more than half of all diet-related deaths and two-thirds of diet-related DALYs [2]. In New Zealand (NZ), the average population diet for both adults and children is low in fruit, vegetables, whole grains, legumes, nuts and seeds with an excess intake of foods high in sodium and sugar [2]. Similar to other high income countries, poor diet is a leading cause of deaths in NZ, accounting for nearly 20% of total deaths in 2017 [2].

Unhealthy food environments contribute to unhealthy diets [3]. Most foods consumed in high income countries are processed or pre-prepared by the food industry [4–7], and associations between adverse health outcomes and high intake of processed foods have been reported [7–10]. Globalization of food production has also led to market domination by companies that produce mainly highly palatable processed foods [11] with low production costs, a long shelf-life, and a high retail value [12]. The packaged food and non-alcoholic beverage industry (referred to from hereon as the packaged food supply) is the world's leading fast-moving consumer goods industry [13], growing 25% between 2011 and 2016, faster than world population growth [14]. In NZ, the packaged food industry is experiencing steady moderate sales growth [13] with many New Zealanders considering price and convenience the most important factors when making decisions around food [15]. Therefore, the packaged food industry has an important role to play in improving population diets and preventing non-communicable chronic diseases [16].

In the United Kingdom [17] and Australia [18], a relatively small number of companies own the majority of brands and products. In NZ in 2018, the five packaged food companies with the highest market share accounted for one-fifth of sales and have maintained steady growth in the local market [13], with numerous brand acquisitions [19], particularly from overseas investors. In contrast to several companies in the NZ packaged food market, the non-alcoholic beverage market is dominated by just two major companies [13]. Therefore, policies targeted to the national packaged food and beverage supply and its contributing companies are an opportunity to improve health.

A recent 2020 assessment of Government food environment policies and infrastructure support [20] concluded that New Zealand is lagging behind other nations in implementing several major policies to improve food environments. NZ lacks clear government leadership and has no guiding strategy. Although New Zealand has a regulatory system for labelling of health and nutrient claims [21] and the New Zealand and Australian governments have implemented a voluntary front-of-pack labelling system (the Australasian Health Star Rating system based on a nutrient profiling model and implemented in 2014) [22], there are no Government-led targets for nutrients of concern for either packaged food or out-of-home meals. However, the Heart Foundation implemented a voluntary food reformulation programme in 2007 [23], funded by the Ministry of Health. The Heart Foundation programme involves working with industry to set and achieve voluntary reformulation targets. To date, forty-eight targets for sodium, saturated fat and/or total sugar have been set across twenty food categories. Nutrient profiling is used to rank the healthiness of individual foods [24]. The Health Star Rating (HSR) is the most commonly used front-of-pack labelling system in NZ and Australia. The frequency of consumption of foods can also be recommended by a country's dietary guidelines, such as the Australian 'Discretionary Food List' [25]. Finally, a more recent classification system is NOVA, which classifies foods and beverages by degree of processing [26]. Ultra-processed foods have been associated with an increased risk of negative diet-related health

outcomes [7–9] including excess weight gain [10]. However, there has been some criticism of NOVA for inconsistency in classification of products and inconsistent associations between consumption and diet-related health outcomes in national surveys [27].

This cross-sectional study represents a comprehensive assessment of major packaged foods and beverages in the NZ food supply. Using data collected from four NZ supermarkets (from four individual brands) in 2018 we aimed to: a) identify and describe the companies that manufactured the packaged food and non-alcoholic beverage products available for sale; b) identify the companies dominating the packaged food supply (making available the majority of products); c) assess the healthiness of all products available for sale and of products sold by dominant companies' by applying three food classification systems: the Australasian HSR front-of-pack nutrition labelling system [28], the Australian Dietary Guidelines core/discretionary classification [25], and the NOVA classification of level of processing [26]; d) compare the number of stars allocated to products which do and do not display the HSR on the front-of-pack and; e) use the findings from b) to c) to assess the potential for food reformulation within selected food sub-categories. This study sought to provide useful information to encourage government to act to improve the packaged food supply and encourage manufacturers to make changes to improve the healthiness of their product portfolios.

## Materials and methods

### NZ supermarkets packaged foods database

The major source of information for this study was Nutritrack, a NZ branded food-composition database developed by the National Institute for Health Innovation at The University of Auckland [29]. Nutritrack is part of an international collaborative approach for collation and sharing of data for packaged and processed foods to enable monitoring of their nutrient composition to support government and food industry efforts to improve the healthfulness of products [30]. Nutritrack is an annual inventory of packaged foods and non-alcoholic beverages available in major NZ supermarkets (herein referred to as packaged foods). The distribution of packaged food in NZ is dominated by supermarkets, accounting for 75% of all purchases [13]. Supermarkets are dominated by two retailers, Woolworths NZ Ltd and Foodstuffs NZ Ltd, which own the four store brands from which Nutritrack data are collected [13]. The database contains information on front-of-pack labels, nutrition and ingredient information as displayed on the package at one point in time each year [29]. The annual surveys are conducted by trained field workers who collect data using a customised smart phone application to take photographs and record barcodes for each food product. All unique products displaying a nutrition information panel [NIP] are included in the surveys. Different pack sizes of the same product have unique barcodes, and each product pack variant is included [29]. Information on products without a NIP, such as unpackaged fruits and vegetables, and fresh meat and alcohol, is not collected, nor is information on vitamins and supplements or seasonal foods. Nutritrack data quality is checked annually and the error rate in 2018 across key fields was <1%. All products sourced are classified in a hierarchical structure of 15 food groups (e.g. bread and bakery products), 66 categories (e.g. biscuits), 201 sub-categories (e.g. sweet filled biscuits) and 443 smaller sub-categories (e.g. chocolate biscuits) [31].

This observational study used information from the Nutritrack 2018 data collection, which was sourced from four NZ supermarket brands (one large store each) between February and April in 2018: New World, Four Square, Countdown and PAKn'SAVE. In total, Nutritrack contained information on 15,192 food products in 2018. Data from all available products in these supermarkets were collected and no sampling process was applied. All collected data were included in the analyses, except for the removal of products with multiple serving sizes or

incomplete data (as described below). These methods are similar to those previously used to assess the healthiness of the food supply in the US [32] and Australia [33].

## Information on packaged food companies

Information about food companies (types, names and brands of products produced) that manufactured the packaged food and beverage products registered for all products in Nutritrack in 2018 was gathered from the product labels, through website searches, by a trade-mark search of the NZ Intellectual Property Office trademarks website [34] or the NZ companies' register [35].

To identify the companies that dominated (sold the majority of) the packaged food and non-alcoholic beverages in the NZ market in 2018, information on market share (% retail value) was retrieved from the *Euromonitor International Passport Global Market Information Database* [13]. The dominant food companies in the NZ market were defined as those with a share of 1% or more in 2018. The major supermarket retailers in NZ (Foodstuffs, Woolworths NZ) were included within the packaged food manufacturers and their market share referred only to the own-brand labels of their respective supermarkets.

## Products inclusion and exclusion criteria

Analyses were conducted for all products in Nutritrack 2018 together, and separately for products manufactured by each of the food companies identified as dominant. The initial Nutritrack 2018 database included 15,192 products. To avoid skewing results, where multiple packs of identical products existed, the product with the smallest pack size only was included in the analysis. After exclusion of multiple pack sizes (n = 744; 4.9%) products from food categories that do not contribute significantly to nutrient intake or are not required to display NIPs were excluded (i.e. baking powders, chewing gum, cough lollies, herbs and spices, plain teas and coffees, yeasts and gelatins, n = 723, 4.8%) (S1 Table). Products with missing NIP information (n = 219; 1.4%) were excluded with the following exceptions: products with no saturated fat data that had ≤ 1g fat per 100g, (assigned '0' for saturated fat); products with no sugar value that contain no carbohydrate (assigned '0' for sugar); and plain water (assigned '0' for nutrients). After the exclusion criteria were applied, 2.8% of the products had missing information for HSR value and were then, excluded from analysis assessing the healthiness of products by this classification system. Fig 1 illustrates the exclusion criteria of products including the number of products analyzed in Nutritrack and within the major food companies according to the three metrics of food healthiness.

## Assessment of product healthiness

There is no international, universally accepted method to classify foods as healthy (or not), with classification systems and nutrient profiling models developed for different purposes and contexts [36]. Therefore, the healthiness of the companies' product portfolios was assessed using three methods or systems, each which classifies foods using a different definition and criteria of healthy:

**Health Star Rating.** Food products were assigned a rating between 0.5 (least healthy) and 5.0 stars (most healthy) in half-star increments based on the nutritional content of the product. This voluntary front-of-pack nutrition labelling system used in NZ and Australia determines healthiness based on a balance of energy, saturated fat, total sugar and sodium per 100g (mandatory to display on the nutrition information panel (NIP)), and with dietary fibre, protein and percentage of fruit, vegetables, nuts and legumes (FVNL) [28]. For products displaying the HSR value on the pack, the declared HSR value was used. For products not displaying the HSR on the pack, HSR was calculated using the HSR algorithm and nutritional information provided on the

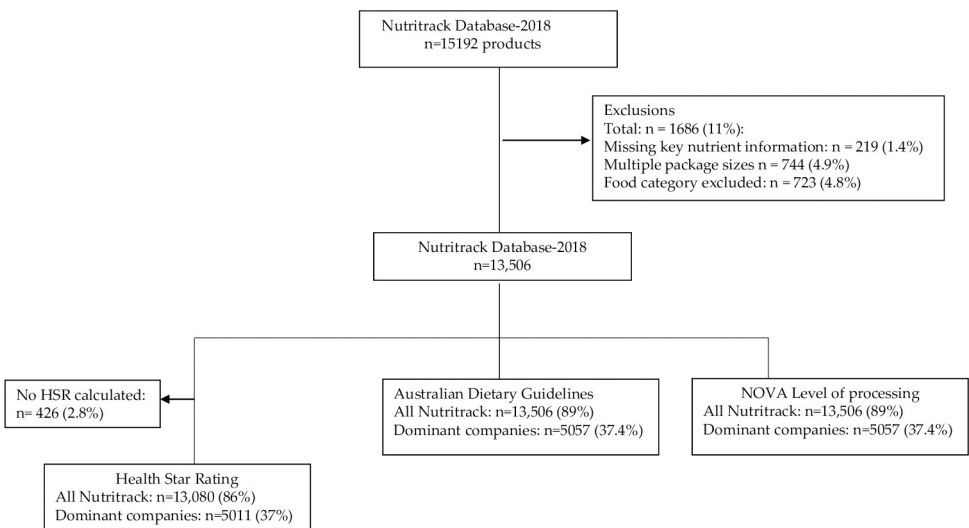

**Fig 1. Flow-chart illustrating the exclusion criteria of products including the number of products analyzed for all products in Nutritrack and within the dominant food companies.**

product's NIP [28]. For some products, information was not available for fibre and FVNL (these are not mandatory to display on the pack of NZ food products [21]); in such cases points for fibre were estimated based on the median value for other products within the category and points for FVNL were assigned per category based on the percentage of FVNL of a sample of products from the category. Estimation of HSR value was not possible for single ingredients products (e.g. vinegar) and for products belonging to food categories with an insufficient number of products with fibre values (<20%) in order to reasonably impute fibre content. Products were classified as 'healthy' if the product had an HSR ≥ 3.5 based on prior research demonstrating that this threshold discriminates between healthy and unhealthy products [37].

**Core/discretionary.** Foods were classified as core or discretionary based on definitions in the 2013 Australian Dietary Guidelines [25]. As the Australian and NZ food supplies and dietary guidelines [21] are similar, it was considered appropriate to use this measure.

**NOVA.** The level of industrial food processing was assigned at a category level where possible according to the NOVA classification framework [26]. Foods and beverages were classified as 'unprocessed or minimally processed foods', 'processed culinary ingredients', 'processed foods' or 'ultra-processed'. Where possible, a level of processing was assigned at the Nutritrack sub-category level. If less than 70% of products in a sub-category were of the same degree of processing, or if the category included minimally processed and ultra-processed products [26], the product was assigned a level of processing at the individual level. Due to heterogeneity of the level of food processing in some food categories, among the 13,506 products with information for NOVA classification, 918 products (6.8%) were classified manually at the individual level using the ingredient list displayed on the label. Principles were developed to guide decisions (S2 Table), particularly regarding the type of ingredients added (such as preservatives, stabilisers) based on a published, comprehensive description of each level of processing from Monteiro et al. [26].

## Potential for food reformulation within specific food sub-categories

The potential for food reformulation was assessed within selected food sub-categories which are considered part of a healthy eating pattern by the NZ Eating and Activity

Guidelines [38] and using the NZ Heart Foundation reformulation targets [23]. The rationale to limit this assessment at the level of food-subcategories instead of at the level of food groups was to be able to compare similar products. At the food group level there is a heterogeneous mix of products and level of healthiness. The selected food sub-categories were breads, breakfast cereals, cheese, yoghurt, margarines, and canned baked beans. For each of these food sub-categories the proportion of products meeting the NZ Heart Foundation reformulation targets for sodium and/or sugar [23] was assessed, within the Nutritrack dataset and according to dominant food companies that had products within the selected food sub-category amongst their portfolios.

## Data analysis and statistics

Descriptive statistics of numbers, proportions, means, standard deviations (SD) and ranges were used to summarize the findings of this observational study. For all products in Nutritrack in 2018 and within products from dominant companies, the mean (SD) HSR and the proportion of products with HSR $\geq$3.5; discretionary and ultra-processed products was calculated. Statistically significant differences in mean HSR and products with HSR $\geq$3.5 for products displaying and not displaying the HSR front-of-pack label was assessed for all products in Nutritrack and for products from the dominant companies. T-tests for independent samples were applied to examine statistically significant differences in mean values and Pearson-chi-square tests were used to examine proportions. Analyses were performed using SPSS software (version 25, IBM SPSS Statistics), and all tests were two-sided at 5% significance level.

## Results

### Number and types of food companies

In total, the 13,506 products from Nutritrack database were matched to 1,767 brands and 1,214 companies. The total number of products per company ranged from one to 992 and the mean number of products per company was 12.5 (+ 50.9). Among the 545 NZ registered companies (almost half of companies represented in Nutritrack), most were NZ Limited companies (n = 375, 30.9% of all companies in Nutritrack) (Table 1). NZ companies with an ultimate holding company (UHC) from another country which has control over all subsidiary

**Table 1. Number and proportion of packaged food companies and mean (SD) and range of products under the different types of food companies, all companies and products in Nutritrack database, 2018.**

| Type of company | Number of companies | Number of products per company | | |
|---|---|---|---|---|
| | N (%) | Mean | Standard Deviation | Range |
| NZ company | 545 (45.0) | 13.1 | 52.5 | 1–992 |
| NZ Limited Company, co-operative, charity | 375 (30.9) | 14.5 | 56.8 | 1–992 |
| NZ Limited Company with NZ UHC | 35 (2.9) | 26.2 | 47.1 | 1–256 |
| NZ Limited Company with Australian UHC | 19 (1.6) | 27.8 | 198.8 | 3–239 |
| NZ Limited Company with UHC in another country (not NZ or Australia) | 39 (3.2) | 85.5 | 135.8 | 1–691 |
| Company not registered, removed from register, or in receivership or liquidation | 77 (6.4) | 5.6 | 4.9 | 1–22 |
| Australian company | 123 (10.1) | 9.3 | 14.5 | 1–113 |
| Overseas company | 546 (44.9) | 4.1 | 5.8 | 1–65 |
| Total | 1214 (100) | 12.5 | 50.9 | 1–992 |

SD, Standard Deviation; NZ, New Zealand; UHC, Ultimate Holding Company.

companies represented 3.2% of all companies. The three top companies with the most products in descending order were: NZ Limited Company with UHC in another country (not NZ or Australia), NZ Limited Company with Australian UHC, and NZ Limited Company with NZ UHC. Despite representing 55% of the companies in Nutritrack, companies registered in Australia or other countries tended to have fewer products per company with a mean of 9.3 and 4.1 products, respectively (Table 1).

Twenty-two food companies dominated the packaged food market in 2018. Nineteen packaged food companies and three non-alcoholic beverage companies held, respectively, 67% and 75% of the NZ market share but just 37% of products within the dataset. Twelve (55%) of the dominant companies were NZ Limited companies with an UHC in a country other than NZ or Australia, six (27%) were NZ Limited companies, a co-operative or a charity, and the remaining four (18%) were Australian linked (NZ Limited company with Australian UHC or an Australian company). The number of brands per company ranged from two (Dairyworks) to 37 (Goodman Fielder). Supermarket retailers had the highest number of products per company with their own-brand private labels (Foodstuffs, n = 904, and Woolworths NZ n = 851) (data not presented in table).

## Health Star Rating, Australian Dietary Guidelines and NOVA classification

Across all products in Nutritrack, 41% were classified as 'healthy' (HSR $\geq$3.5), 53% were classified as discretionary and 69% as ultra-processed. This was similar for the 22 dominant companies, with 44% of products classified as 'healthy', 50% as discretionary and 74% as ultra-processed.

Table 2 presents the healthiness of the portfolios of the 22 dominant food companies using the three classification systems. The healthiness of product portfolios varied considerably among manufacturers. The range in mean HSR ($\pm$SD) was 1.0 +0.8 (Mondelēz) to 4.1 $\pm$0.7 (Sanitarium), and the percentage of products classified with an HSR $\geq$3.5 ranged from 2.7% (Mondelēz) to 95.5% (Sealord). The top three companies with the highest mean HSR were Sanitarium, McCain Foods, and Sealord, with portfolios of at least 69.8% of products with an HSR $\geq$3.5; over two-thirds of their products were classified as core. Of note was that McCain Foods had 13% of products missing information for HSR, and thus there is a higher potential for overestimation of the mean HSR and proportion of products with HSR $\geq$3.5 in this company's portfolio.

All non-alcoholic beverage companies (Coca-Cola, Frucor Suntory, The Better Drinks Co) had a low mean HSR with at least 70% of their products with an HSR <3.5 and at least 83% of their products classified as ultra-processed (Table 2). Among these three companies Cola-Cola performed worst (mean HSR 1.8; HSR $\geq$3.5 = 8.0%; discretionary products = 78.4%; ultra-processed products = 96.3%).

The proportion of discretionary foods in the product portfolio of the dominant companies ranged from 1.1% (Sealord) to 100% (Bluebird Foods)). Sixteen companies (72.7%) had product portfolios that contained at least 30% of discretionary products. The range of variation of ultra-processed products within companies'portfolios was 11% (Dairyworks) to 100% (Arnott's, Griffin's Foods and Bluebird Foods). Half of the companies (n = 11) had portfolios that contained more than 90% of ultra-processed products and 18 companies (81.8%) had product portfolios with more than 70% of products ultra-processed (Table 2). Healthiness by manufacturer is highly dependent on product portfolio and differences in products available for sale within sub-categories can lead to large differences in relative healthiness. Both George Weston Foods and Griffin's Foods predominantly sell bread and bakery products but 86% of George Weston Food's products in this food group are bread and all of Griffins Foods

**Table 2. Top food groups manufactured by the dominant companies and assessment of healthiness of their product portfolios according to three classification systems, Nutritrack database, 2018.**

| Company | Top food groups per company (up to 3) | Number of products assessed | Food classification system indicating healthiness | | | | |
|---|---|---|---|---|---|---|---|
| | | | Health Star Rating [28] * | | | Dietary Guidelines [25] | Level of processing [26] |
| | | | Mean | SD | Proportion HSR ≥ 3.5 (%) | Proportion discretionary (%) | Proportion ultra-processed (%) |
| Arnott's | Bread and bakery products; Convenience foods; Sauces, dressings, spreads and dips | 117 | 2.2 | 1.3 | 32.5% | 70.1% | 100.0% |
| Bluebird Foods | Snack foods | 66 | 1.9 | 0.8 | 6.1% | 100.0% | 100.0% |
| Dairyworks | Dairy | 64 | 2.7 | 1.1 | 40.6% | 14.1% | 10.9% |
| Fonterra | Dairy; Edible oils and fats | 326 | 2.8 | 1.5 | 40.2%* | 30.8% | 57.7% |
| Foodstuffs (own brand) | Packaged fruit and vegetables; Bread and bakery products; Cereal and cereal products | 904 | 3.0 | 1.3 | 50.9% | 45.5% | 59.0% |
| George Weston Foods | Bread and bakery products; Meat and meat products | 94 | 3.4 | 0.8 | 67.0% | 27.7% | 98.9% |
| Goodman Fielder | Bread and bakery products; Dairy; Sauces, dressings, spreads and dips | 427 | 2.9 | 1.3 | 49.1%* | 27.6% | 71.7% |
| Griffin's Foods | Bread and bakery products; Cereal and cereal products | 134 | 1.9 | 1.2 | 14.9% | 93.3% | 100.0% |
| Heinz Wattie's | Packaged fruit and vegetables; Convenience foods; Sauces, dressings, spreads and dips | 564 | 3.3 | 1.2 | 60.6%* | 47.0% | 72.3% |
| Hellers | Packaged meat and meat products | 101 | 1.7 | 0.7 | 6.9% | 87.1% | 84.2% |
| Kellogg's | Cereal and cereal products; Snack foods | 55 | 3.1 | 1.3 | 54.5% | 41.8% | 98.2% |
| McCain Foods | Convenience foods; Fruit and vegetables | 61 | 3.9* | 0.9 | 69.8%* | 34.4% | 72.1% |
| Mars | Sauces, dressings, spreads and dips; Confectionary; Cereal and cereal products | 135 | 2.2 | 1.3 | 31.9% | 78.5% | 82.2% |
| Mondelēz | Confectionery; Dairy; Bread and bakery products | 185 | 1.0 | 0.8 | 2.7% | 89.7% | 96.5% |
| Nestlé NZ | Sauces, dressings, spreads and dips; Confectionery; Cereal and grain products | 257 | 2.5 | 1.4 | 40.9%* | 72.4% | 93.0% |
| Sanitarium | Cereal and grain products; Sauces, dressings, spreads and dips | 81 | 4.1 | 0.7 | 87.7% | 11.1% | 91.4% |
| Sealord | Fish and seafood products | 89 | 3.8 | 0.3 | 95.5% | 1.1% | 71.9% |
| Unilever NZ | Dairy; Sauces, dressings, spreads and dips; Convenience Foods | 153 | 2.4 | 1.1 | 27.6%* | 58.8% | 96.1% |
| Woolworths NZ (own brand) | Packaged fruit and vegetables; Bread and bakery products; Cereal and cereal products | 851 | 3.0 | 1.4 | 49.0%* | 45.7% | 59.0% |
| Coca-Cola | Non-alcoholic beverages | 173 | 1.8* | 1.0 | 8.8%* | 78.4% | 96.0% |
| Frucor Suntory | Non-alcoholic beverages | 167 | 2.6* | 1.0 | 30.3%* | 51.9% | 93.8% |
| The Better Drinks Co | Non-alcoholic beverages | 53 | 2.7* | 1.6 | 30.8%* | 32.7% | 82.7% |

HSR, Health Star Rating.

* Percentage (%) of products with missing information within company: McCain Foods 13.1% (frozen pizzas), Coca-Cola 3.5% (sparkling water), Frucor Suntory 3.0% (sparkling water), Fonterra (1.5%), Nestlé NZ (1.2%), Heinz Wattie's 1.1%, Mondelēz 1.1%, Woolworths NZ 0.7%, Unilever NZ 0.7%, Goodman Fielder 0.2%.

products in this food group are biscuits. Hence the differences in overall HSR are considerable (mean 3.4, 67% ≥3.5; mean 1.9, 14.9% ≥3.5 respectively).

Companies that have a wide portfolio of food categories tended to have a low mean HSR and a high proportion of products classified as discretionary foods and ultra-processed. For example, Nestlé NZ (mean HSR 2.5, 72.4% discretionary, 93.4% ultra-processed), and Unilever

NZ (2.4, 58.8%, 96.1%), Mars (2.2, 78.5%, 82.2%). The products of the two supermarket retailers, Woolworths NZ and Foodstuffs, were their own-brand products, which also span a wide range of food categories. In contrast, the product portfolios of the retailers performed better than the brand leaders named above with diverse portfolios, shown by higher mean HSRs (both retailers mean of 3.0) and a lower proportion of products that were discretionary (both retailers 46%) or ultra-processed products (59% and 60% respectively).

## Health Star Rating displayed/not displayed on front of pack

Across all Nutritrack products, almost eight in 10 (79%) did not display the HSR label on the pack and the mean HSR (SD) of products displaying the HSR label was higher than the mean HSR of products not displaying the HSR label on pack (3.2+ 1.3 *versus* 2.5+ 1.4, t = 23.818; p = 0.000). There was also a higher proportion of products with an HSR ≥3.5 among products displaying the HSR in relation to products not displaying the HSR label (58.6% versus 36.2%, chi-square = 454.732; p = 0.000). Similar findings were observed for products sold by the dominant food companies, where products displaying HSR had a mean HSR of 3.1 (+1.4) and products not displaying HSR had a mean HSR of 2.7 (+1.3) (t = 10.577; p = 0.000). Fifty-five percent of products displaying HSR had an HSR ≥3.5 compared to 41% of products not displaying the HSR label (chi square = 87.743; p = 0.000) (data not presented in table).

Dominant companies with the highest percentage of products that displayed the HSR on front-of-pack were: Sanitarium (90%), Kellogg's (73%) and Foodstuffs (own-brand) (73%). The dominant companies with the lowest percentage of products displaying the HSR label were Goodman Fielder (3%), Hellers (3%) and George Weston Foods (6%). Six companies did not display the HSR on any products (Mars, Dairyworks, The Better Drinks Co, Griffin's Foods, Bluebird Foods, and Mondelēz).

Fig 2 visually displays the proportion of products within each HSR increment that did, or did not, display the HSR on the label. Patterns of HSR display differ across companies. Foodstuffs and Woolworths display the HSR on products across the range of HSR values while companies like Fonterra and Heinz Watties and Nestlé NZ display the HSR mainly on products with a higher HSR.

## Potential for reformulation within selected food sub-categories

Table 3 indicates the proportion of products meeting Heart Foundation sodium and sugar targets for selected food sub-categories, both within Nutritrack and for individual dominant companies. Within all products from Nutritrack, the Heart Foundation targets for sodium were met by the majority of products in three of the five sub-categories assessed; breakfast cereals (92% of products) margarine (90%), and cheese (72%). For the remaining two categories (bread, baked beans), less than half of products met the Heart Foundation sodium targets. Among products from the portfolios of the dominant food companies displayed in Table 3, 36% of products from dominant companies met the sodium target for bread compared to 43% of all companies, and for baked beans 21% of products from the dominant companies met the target compared to 33% of all companies. One of the major bread manufacturers had 28% of products meeting the target and one of the baked beans manufacturers had no products meeting the sodium target.

For sugar, the Heart Foundation targets were met by the majority of products in one of the three sub-categories assessed: breakfast cereals (77% of products). For the remaining two categories (yoghurt and baked beans), less than half of products met the Heart Foundation sugar targets. One-third (33%) of products of the dominant companies met the sugar target for baked beans compared to 40% to all products in Nutritrack. For yoghurt, 40% of the products

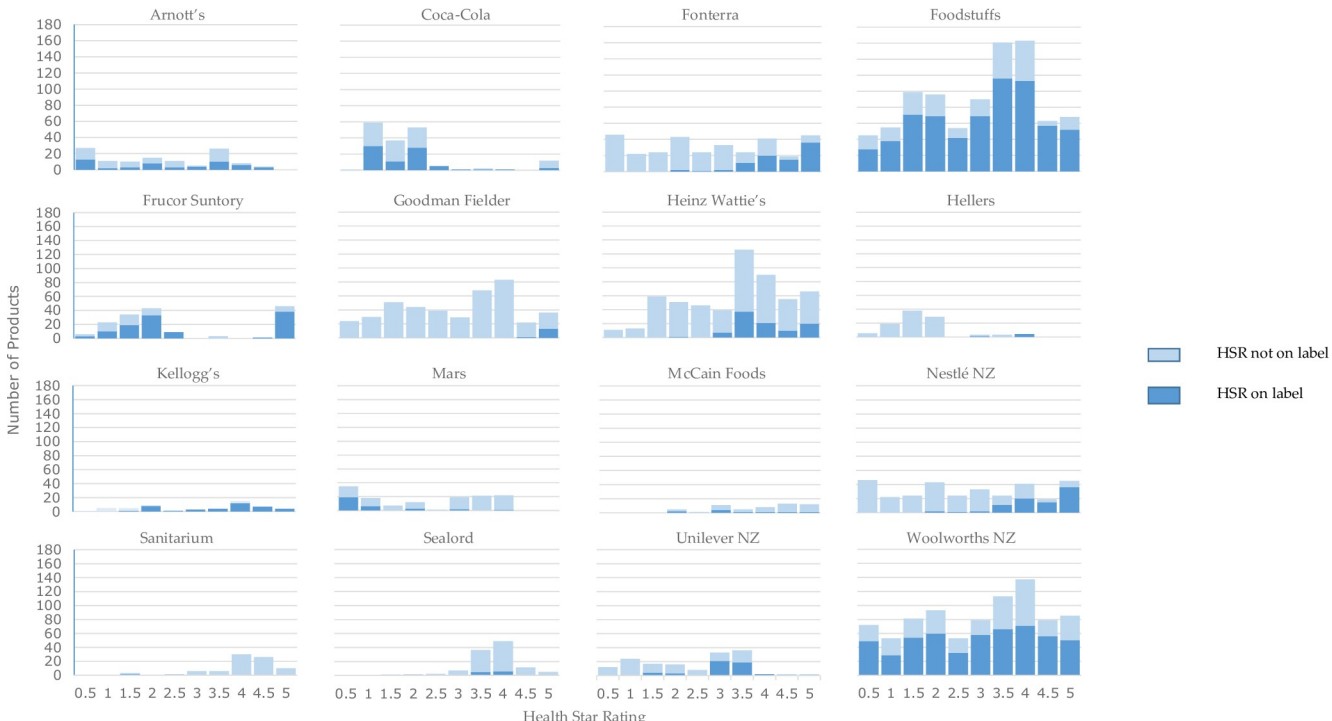

**Fig 2. Visual comparison of number of products with HSR displayed and not displayed on the pack by food company and according to HSR value, Nutritrack database, 2018. N = 16 food companies. Six food companies are not represented because HSR was not displayed on any products within portfolios.**

of the dominant companies met the target compared to 47% of all products in Nutritrack. These findings clearly demonstrate that it would be feasible to reduce sugar and sodium of the selected sub-categories across all food company portfolios.

## Discussion

### Main findings compared with other studies

The packaged food supply in NZ supermarkets is generally unhealthy with products receiving a mean HSR of 2.7 out of 5.0, 53% considered discretionary and 69% of products considered ultra-processed. These findings are similar to recent cross-sectional studies of the Australian [32] and the US [33] packaged food supplies using similar methodology, measures of healthfulness and level of processing. Although the Health Star Rating system is only present on foods in Australasia, this system was used in the US study as the system underpinning it is the basis for several similar systems used globally [33], including the UK Food Standards Agency Nutrient Profiling Model [39]. Results have been reported for NZ in detail along with mean energy, saturated fat, sodium and total sugar per company [40]. The authors of these previous studies concluded that the respective food supplies were dominated by highly processed packaged food and beverage products, and the mean (SD) HSR was similar to NZ 2.7 (1.4) compared with the US 2.7 (1.4) and Australia 2.8 (1.4). A similar number of products was classified as ultra-processed in NZ (69%) and the US (71%) with a lower proportion in Australia (61%). This is of concern as diet-related risk factors (including high salt intake, high saturated fat intake) and excess energy account for 28.6% of health loss in NZ [41]. Though there are over 15,000 unique packaged food and beverage items with NIPs for sale in major NZ

**Table 3. Percentage of selected food sub-categories meeting the Heart Foundation reformulation targets for sodium or sugar in 2018, for all products in Nutritrack and within products from companies' portfolios.**

| Selected food sub-categories | Products meeting Heart Foundation target [23] | | Heart Foundation target Maximum level per 100g [23] |
|---|---|---|---|
| | Sodium n (%) | Sugar n (%) | |
| Bread | | | Sodium |
| All products in Nutritrack | 125 (42.7) | -- | Leavened bread 380mg |
| Portfolio—Foodstuffs (own-brand) | 7 (35.0) | -- | Unleavened bread 450mg |
| Portfolio—George Weston Foods | 14 (28.0) | -- | |
| Portfolio—Goodman Fielder | 27 (42.9) | -- | |
| Portfolio—Woolworths NZ (own-brand) | 9 (64.3) | -- | |
| Breakfast cereal | | | Sodium |
| All products in Nutritrack | 339 (92.1) | 283 (76.9) | Puffed rice and cornflakes 500mg |
| Portfolio—Foodstuffs (own-brand) | 23 (95.8) | 13 (54.1) | Oat based muesli, porridge 200mg |
| Portfolio—Goodman Fielder | 16 (93.8) | 12 (75.0) | Breakfast biscuits 300mg |
| Portfolio—Kellogg's | 35 (85.4) | 17 (41.5) | Other ready-to-eat cereals 400mg |
| Portfolio—Nestlé NZ | 28 (100) | 17 (60.7) | Sugar |
| Portfolio—Sanitarium | 28 (77.8) | 34 (94.4) | All cereals: 22.5g |
| Portfolio—Woolworths NZ (own-brand) | 21 (100) | 15 (71.4) | |
| Cheese (Cheddar, processed, mozzarella) | | | Sodium |
| All products in Nutritrack | 196 (72.1) | -- | Cheddar and cheddar-style 710mg |
| Portfolio—Dairyworks | 40 (78.4) | -- | Mozzarella 550mg |
| Portfolio—Fonterra | 38 (61.3) | -- | Processed 1270mg |
| Portfolio—Foodstuffs (own-brand) | 17 (77.3) | -- | |
| Portfolio—Goodman Fielder | 14 (60.9) | -- | |
| Portfolio—Woolworths NZ (own-brand) | 33 (80.5) | -- | |
| Yoghurt | | | Sugar 8.5g |
| All products in Nutritrack | -- | 95 (46.8) | |
| Portfolio—Fonterra | -- | 28 (45.9) | |
| Portfolio—Goodman Fielder | -- | 12 (31.6) | |
| Canned baked beans in tomato sauce | | | Sodium 350mg |
| All products in Nutritrack | 10 (33.3%) | 12 (40.0) | Total sugar 5g |
| Portfolio—Foodstuffs (own-brand) | 4 (0%) | 2 (50.0) | |
| Portfolio—Heinz Wattie's | 5 (29.4%) | 5 (29.4) | |
| Portfolio—Woolworths NZ (own-brand) | 0 (0%) | 1 (33.0) | |
| Margarine (oil-based) | | | Sodium 400mg |
| All products in Nutritrack | 35 (89.7) | -- | |
| Portfolio—Foodstuffs (own-brand) | 6 (100) | -- | |
| Portfolio—Goodman Fielder | 6 (100) | -- | |
| Portfolio—Unilever NZ | 11 (91.7) | -- | |

supermarkets, there are also a considerable number of healthier options available such as fresh fruit and vegetables.

The HSR was only displayed on 21% of all products in Nutritrack and 39% of products produced by the dominant companies. Some companies (Sanitarium, Foodstuffs, Kellogg's) had the HSR displayed on the majority of products while others did not display the HSR on any products. For some companies the HSR was displayed on products across the range of HSR

values (for example, Woolworths NZ, Foodstuffs) but for other companies (for example, Heinz Wattie's and Nestlé NZ) the HSR appeared selectively displayed on healthier products, such as packaged fruit and vegetables and hot cereals. The mean HSR of products displaying the HSR was higher than products not displaying the HSR, across all products in Nutritrack and within the products of the dominant companies. Similarly, an analysis using NZ Nutri-track data conducted in 2016 found 84% of products that displayed the HSR had higher (healthier) ratings of 3.0 to 5.0 stars [42]. If the HSR was to be made mandatory, then all relevant products would display the nutrition label enabling consumers to make better-informed choices when selecting packaged food and beverages.

There is a proliferation of packaged foods and brands in the NZ market with steady growth [13], though market share is dominated by a relatively small number of companies, especially in the non-alcoholic beverage retail market, which is heavily dominated by two major companies (Coca-Cola and Frucor Suntory). Most of the companies with the highest market share are not NZ owned (16 of the major 22 companies) which can make it more challenging for policy-makers and public health advocates to influence decisions on product portfolio, reformulation and promotion.

There are differences in the classification of healthiness by indicator so using multiple systems to monitor healthiness is a strength of this study. This analysis used three systems to classify food as healthy, each with a different principle of defining healthy food thus providing a comprehensive picture of the NZ packaged food supply. The HSR considers the actual amounts of six nutrients and positive food components, the proportion of discretionary foods is guided by the food group recommendations of the Australian Dietary Guidelines, and the proportion of ultra-processed products depends on the degree and method of processing along with added ingredients. Therefore, some manufacturers can have a high mean HSR, a low proportion of discretionary foods, and yet a high proportion of ultra-processed foods; for example, Sanitarium (breakfast cereals), and George Weston Foods (bread). Conversely, most of the products of Dairyworks are core and not ultra-processed yet only two-fifths have an HSR ≥3.5 due to the saturated fat content of many hard cheeses.

Within categories of commonly consumed foods, such as bread, breakfast cereals and dairy, there are many companies, brands and products in the NZ market but only a few dominant players despite the breadth of manufacturers. Efforts to improve the healthiness of products, and promotion of healthier options should be directed at dominant companies because of their high market share. However, most of the dominant companies are overseas owned due to globalization of food production [11], indicating there is considerable overlap among manufacturers who sell foods globally, therefore potential exists for healthier products to be manufactured and sold widely to make a large difference to global population intakes, particularly for sodium. Though products may differ between countries according to local regulations and preferences, ongoing monitoring of the global food supply by individual countries using similar methodology and assessment tools could assist policymakers to prioritise policy actions for improving the food supply to reduce the impact of diet-related ill-health [30].

## Implications

To maximise adherence to national dietary guidelines, consumers need a food environment that enables them to follow an eating pattern based mostly on whole foods and less processed foods that are low in added sugar and sodium and leads to improved health outcomes [1, 7–10]. While many of the packaged foods and beverages available in NZ are excessively energy dense and high in sodium, saturated fat and sugar [40] some companies do provide healthier and less processed products. A range of solutions is required to ensure the food supply is

acceptable, affordable and accessible to reduce the exposure and availability of unhealthy foods and encourage consumers to choose the healthier items on offer. Such solutions include the setting of government-led reformulation targets for processed foods, monitoring population intake targets for nutrients of concern such as sodium and sugar, mandatory front-of-pack labelling and a substantial tax on sugary drinks. These are all actions that were prioritised by a public health expert panel that assessed Government progress on food environment policies and prioritised actions in NZ [20]. For reformulation, the Panel recommended a two-tier system for reducing sodium and added sugar in key food categories i.e. (1) setting mandatory maximum levels, similar to the voluntary Heart Foundation targets, to remove the products with the highest levels of sodium and added sugar in product categories, and (2) setting and monitoring targets for voluntary reductions in sales-weighted averages in key food groups as a collective target with the aim of reducing the average sodium or sugar content in a category.

Front-of-pack labelling, such as the Health Star Rating system in Australia and New Zealand can encourage reformulation [42]. However, uptake has been slow with the HSR on only one-fifth of products in New Zealand since its introduction in 2014 and with the HSR selectively displayed on products with a higher HSR, therefore public health experts recommend it should now be mandatory. However, a recommendation from a recent five year review was for the HSR system to continue as a voluntary system with the addition of industry uptake targets [43].

Reformulation can contribute to healthier eating by improving the healthiness of packaged foods, particularly by reducing the amount of sodium added. While the NZ government has not established population intake targets, or food reformulation targets for sodium and sugar, the Heart Foundation [23] has voluntary reformulation targets for twenty food categories, with sodium targets for seventeen categories, and total sugar targets for nine. This programme was developed in 2007 initially to reduce sodium levels across packaged foods with a focus on lower cost and higher volume products. While for some categories, a high proportion of products made by the dominant companies met Heart Foundation targets for sodium and/or sugar (margarine, breakfast cereals, cheese), there is a great deal of room for improvement in other categories, particularly bread, baked beans and yoghurt.

The results of this study could stimulate modelling studies such as assessing the impact of reducing the availability of highly processed foods on the health of populations. Ideally, future assessments of the packaged food supply would be complemented with sales data to provide useful information on consumer purchasing patterns in relation to the healthiness of the food supply. A comparison of estimates of nutrient exposure from a store survey of packaged foods with those from household panel food purchases in NZ found that store survey data provided a reasonable estimate of average population exposure to key nutrients from packaged foods [44].

## Strengths and limitations

This study is the first in New Zealand to assess the healthiness of the packaged food supply by company. A strength of the study is that the packaged food data in Nutritrack are collected annually using a standardised approach since 2013, with a large range of products captured. The data are representative of what is on the shelves of the sampled stores during the survey period, though do not represent every packaged food and beverage available in every store throughout the year [29]. It is estimated that 75% of unique products are represented [13]. Further, supermarkets are where most NZ households (87%) purchase foods and drinks weekly or more often [45]. Nutritrack data collection and analysis are independent of interested parties, the food industry.

However, this study is limited by the changing food supply and dependence on the accuracy of the information provided on food packaging. The analyses rely upon the accuracy of data reported on pack by manufacturers, such as nutrient content, and since some manufacturers

use the government-supplied food calculator [28] the true composition of foods may differ from that reported on the product label. For packaged products that did not display the HSR, imputation of some metrics was required for the calculation of the HSR, such as percentage of fruit, vegetable, nut and legume content. Therefore, the HSR may be over or underestimated for a small number of products.

## Conclusions

In 2018, packaged foods and beverages available in NZ supermarkets included in the Nutritrack database were dominated by discretionary products (53%), products with HSR <3.5 (59%) and ultra-processed products (69%). In addition, 79% of products available for sale did not display the HSR, with the HSR being selectively displayed on healthier products i.e. those with higher HSR values ($\geq$3.5). In contrast, some companies had a high proportion of products with an HSR of at least 3.5, commonly displayed the HSR and had committed to reformulation. Dietary guidelines emphasize the importance of whole foods for health, but as packaged foods are prolific in the food supply there is a need to change the proportion of less healthy options available within product portfolios. Another option is reformulation of products and this paper shows that, based on the analysis of five specific food sub-categories, reformulation to reduce sodium and salt is feasible, at least for some products. Efforts to improve the healthiness of products should be directed to the 22 food companies dominating this market share, particularly in the core foods groups which are currently less likely to meet Heart Foundation targets (bread, breakfast cereals, cheese, canned baked beans, yoghurt). Currently, there is no government-led structured reformulation programme in NZ and the front-of-pack labelling system is voluntary. Therefore, government leadership is required to make substantive gains across the food supply and address sub-optimal nutrient intake by setting reformulation targets for core food groups and by mandating the HSR on products. Regular monitoring of the healthiness of the food supply is required to monitor changes in food company policy and product portfolios.

## Supporting information

**S1 Table. Food categories excluded.**
(DOCX)

**S2 Table. Classifying foods in Nutritrack according to level of processing (NOVA).**
(DOCX)

## Author Contributions

**Conceptualization:** Sally Mackay, Helen Eyles, Cliona Ni Mhurchu, Boyd Swinburn.

**Formal analysis:** Sally Mackay, Teresa Gontijo de Castro, Leanne Young.

**Methodology:** Sally Mackay, Helen Eyles, Teresa Gontijo de Castro, Leanne Young.

**Writing – original draft:** Sally Mackay.

**Writing – review & editing:** Sally Mackay, Helen Eyles, Teresa Gontijo de Castro, Leanne Young, Cliona Ni Mhurchu, Boyd Swinburn.

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
