## [Decision Letter · Decision Letter 0]

3 Sep 2020

PONE-D-20-17346

Which companies dominate the packaged food supply of New Zealand and how healthy are their products?

PLOS ONE

Dear Dr. Mackay,

Thank you for submitting your manuscript to PLOS ONE. After careful consideration, we feel that it has merit but does not fully meet PLOS ONE’s publication criteria as it currently stands. Therefore, we invite you to submit a revised version of the manuscript that addresses the points raised during the review process.

We look forward to receiving your revised manuscript.

Kind regards,

Zhifeng Gao

Academic Editor

PLOS ONE

Journal Requirements:

2. As the Nutritrack database is based on data collected from only 4  stores, please consider adding this consideration in your Limitations section.

3. Please provide more information regarding the choice to calculate the potential for food reformulation based solely on the  NZ Heart Foundation reformulation targets.

4. We note that Figure S1Fig in your submission contain copyrighted images. All PLOS content is published under the Creative Commons Attribution License (CC BY 4.0), which means that the manuscript, images, and Supporting Information files will be freely available online, and any third party is permitted to access, download, copy, distribute, and use these materials in any way, even commercially, with proper attribution. For more information, see our copyright guidelines: http://journals.plos.org/plosone/s/licenses-and-copyright.

4. 1.         You may seek permission from the original copyright holder of S1Fig to publish the content specifically under the CC BY 4.0 license.

4.2.    If you are unable to obtain permission from the original copyright holder to publish these figures under the CC BY 4.0 license or if the copyright holder’s requirements are incompatible with the CC BY 4.0 license, please either i) remove the figure or ii) supply a replacement figure that complies with the CC BY 4.0 license. Please check copyright information on all replacement figures and update the figure caption with source information. If applicable, please specify in the figure caption text when a figure is similar but not identical to the original image and is therefore for illustrative purposes only.

Reviewers' comments:

Reviewer's Responses to Questions

**Comments to the Author**

1. Is the manuscript technically sound, and do the data support the conclusions?

Reviewer #1: Partly

Reviewer #2: Yes

Reviewer #3: Yes

2. Has the statistical analysis been performed appropriately and rigorously? 

Reviewer #1: Yes

Reviewer #2: Yes

Reviewer #3: Yes

3. Have the authors made all data underlying the findings in their manuscript fully available?

Reviewer #1: Yes

Reviewer #2: Yes

Reviewer #3: Yes

4. Is the manuscript presented in an intelligible fashion and written in standard English?

Reviewer #1: Yes

Reviewer #2: Yes

Reviewer #3: Yes

5. Review Comments to the Author

Reviewer #1: The results are based on 23 market share of products available for sale in which there were 22 dominating companies producing 31% of products and 17% of brands. The limitations as noted by the authors this study include the changing food supply and dependence on the accuracy of the information provided on food packaging and the varying time the products are on the supermarket shelves. The major source of information for this study was Nutritrack, a NZ branded food composition database developed by the National Institute for Health Innovation at The University of Auckland.

Descriptive statistics of numbers, proportions, means, standard deviations (SD) and ranges were used to summarize the findings. T-tests for independent samples were applied to examine statistically significant differences in mean values and Pearson-chi square tests were used to examine proportions.

There are several issues:

1. There is no real statistical design to the study in terms of the adequacy of the sample. Thus if one can accept the accuracy of the sample with meaningful statistical precision or reasonable margin of error, given the fact that the study is basically descriptive, then the results are very simply stated. The investigators should address this issue.

2. The page numbering is confusing. The manuscript goes to page 21 and then the tables start on page 1 and the ‘Discussion’ section starts on page 3 or 4. Please correct this.

Reviewer #2: This is a New Zealand-specific study that describes the healthfulness of processed retail foods by food category and manufacturer. The summary statistics are adequately presented and described. Although this study is not a modeling study, the data it presents can be used by researchers to motivate modeling studies that explain, for example, why some consumers chose highly processed foods and why some manufacturers choose to reformulate while others do not.

One important limitation of this study that has not been acknowledged is that the reported product and manufacturer market shares are all based on product counts. A more conventional and informative approach is to weight products by sales volume. This requires retail scanner data that should be be available from retailers, if not already from Nielsen. There are likely to be large differences between the weighted and unweighted the summary statistics. For example, Coke Classic is counted equally as Foxton Fizz even though the two have very different sales volume.

Reviewer #3: This paper examines which companies dominate the packaged food supply of New Zealand and how healthy their products are using three different metrics. It is a descriptive paper that contains some new and interesting information. My comments are not substantial:

1. Line 67, what kinds of policies are the authors discussing here. Are these labeling, nutrition info disclosing, reformulation etc? It will be desirable if the authors provide some more details here plus some policy background in NZ, which could add a bit to the motivation of the study. As I read more into the end of the paper, I realize the product reformulation is the main policy angle. Still a bit background on New Zealand related food policy development will be helpful (in the context of international comparison as well).

2. Line 367, I like the international comparison but does the US use the HSR system? A bit clarification is needed here.

3. Related to my point 1, the policy section seems a bit weak in terms of implications.

6. PLOS authors have the option to publish the peer review history of their article (what does this mean?). If published, this will include your full peer review and any attached files.

Reviewer #1: No

Reviewer #2: No

Reviewer #3: No

---

## [Author Response · Author response to Decision Letter 0]

24 Sep 2020

Auckland, 26th September 2020

PLOS ONE- PONE-D-20-17346

Which companies dominate the packaged food supply of New Zealand and how healthy are their products?

Dear Editor and reviewers,

Thank you for the opportunity to review and improve our manuscript. This letter outlines the response to each point raised by the editor and reviewers and indicates the line numbers in the text the amendments can be found (in the version with track changes visible). The reviewer comments are in italics, our response is in standard text with amended text indicated in red. We have submitted a marked-up copy of the manuscript that highlights changes made to the original version and an unmarked version of the revised manuscript without tracked changes.

Kind regards,

Dr Sally Mackay

Research Fellow

University of Auckland

On behalf of the authors

 

Journal Requirements:

Response: This version has been formatted according to PlosOne style requirements.

2. As the Nutritrack database is based on data collected from only 4 stores, please consider adding this consideration in your Limitations section.

Response: We believe that the information we have provided in the methods section (NZ supermarkets packaged foods database) reiterates that Nutritrack encompasses the majority of the packaged foods purchased in NZ. The distribution of packaged foods in NZ is dominated by supermarkets, accounting for 75% of all purchases [*]. Supermarkets in NZ are dominated by two retailers, Woolworths NZ Ltd and Foodstuffs NZ Ltd, which own the four store brands from which Nutritrack data are collected [*]. Nutritrack is an annual inventory of packaged foods and non-alcoholic beverages available in NZ supermarkets. Nutritrack is part of an international collaborative approach for collation and sharing of data from supermarkets to enable monitoring of the nutrient composition of processed foods and to support government and food industry efforts to improve the nutrient composition of products [**]. 

* Euromonitor International. Passport Global Market Information Database. 2018.

** The Food Monitoring Group. Progress with a global branded food composition database. Food Chem. 2013;140:451–457. 

The introduction and the discussion states that Nutritrack encompasses the majority of the packaged foods sold in NZ in 2018.

Lines 98- 99: Using data collected from four NZ supermarkets (from four individual brands) in 2018 

Lines 528-531: The data are representative of what is on the shelves of the sampled stores during the survey period, though do not represent every packaged food and beverage available in every store throughout the year [29]. It is estimated that 75% of unique products are represented [13].

3. Please provide more information regarding the choice to calculate the potential for food reformulation based solely on the NZ Heart Foundation reformulation targets.

Response: We agree that additional information on the choice of the Heart Foundation targets is useful. We have added information about the targets to the revised version of the manuscript in the introduction. New Zealand does not have reformulation targets set by government. The Heart Foundation HeartSafe programme has been in place for 13 years and works closely with manufacturers to set and attain the targets. There are no other New Zealand specific food reformulation targets. 

Line 79-85: there are no Government-led targets for nutrients of concern for either packaged food or out-of-home meals. However, the Heart Foundation implemented a voluntary food reformulation programme in 2007 [23], funded by the Ministry of Health. The Heart Foundation programme involves working with industry to set and achieve voluntary reformulation targets. To date, forty-eight targets for sodium, saturated fat and/or total sugar have been set across twenty food categories. 

4. We note that Figure S1Fig in your submission contain copyrighted images. All PLOS content is published under the Creative Commons Attribution License (CC BY 4.0), which means that the manuscript, images, and Supporting Information files will be freely available online, and any third party is permitted to access, download, copy, distribute, and use these materials in any way, even commercially, with proper attribution. For more information, see our copyright guidelines: http://journals.plos.org/plosone/s/licenses-and-copyright.We require you to either (1) present written permission from the copyright holder to publish these figures specifically under the CC BY 4.0 license, or (2) remove the figures from your submission:

Response: This is a supplementary figure and not essential for the manuscript. Thus, we decided to remove the visual maps of brand from this revised version. Line 64 of the introduction and lines 534-536 of the discussion has been amended to remove reference to visual maps of brands.

Reviewers' comments:

Review Comments to the Author

Reviewer #1: The results are based on 23 market share of products available for sale in which there were 22 dominating companies producing 31% of products and 17% of brands. The limitations as noted by the authors this study include the changing food supply and dependence on the accuracy of the information provided on food packaging and the varying time the products are on the supermarket shelves. The major source of information for this study was Nutritrack, a NZ branded food composition database developed by the National Institute for Health Innovation at The University of Auckland. Descriptive statistics of numbers, proportions, means, standard deviations (SD) and ranges were used to summarize the findings. T-tests for independent samples were applied to examine statistically significant differences in mean values and Pearson-chi square tests were used to examine proportions.

There are several issues:

5. There is no real statistical design to the study in terms of the adequacy of the sample. Thus if one can accept the accuracy of the sample with meaningful statistical precision or reasonable margin of error, given the fact that the study is basically descriptive, then the results are very simply stated. The investigators should address this issue.

Response: As the reviewer indicates, the study is descriptive and we consider descriptive statistics sufficient to answer the study’s research questions. No sampling process was required as we assessed the majority of packaged food products available for sale in NZ supermarkets in 2018 and included all products in our analyses. The presentation of margin errors (such as SD) are appropriate and were also used in a similar analysis of the US food supply by Baldridge et al.

Baldridge A, Huffman M, Taylor F, Xavier D, Bright B, Van Horn L, et al. The Healthfulness of the US packaged food and beverage supply: A cross-sectional study. Nutrients. 2019;11:1704.

We have added wording in the manuscript to be clear that this is an observational study (line 139 & 240) and the wording below in the introduction.

Lines 142-146: Data from all available products in these supermarkets were collected and no sampling process was applied. All collected data were included in the analyses, except for the removal of products with multiple serving sizes or incomplete data (as described below). These methods are similar to those previously used to assess the healthiness of the food supply in the US [32] and Australia [33].

6. The page numbering is confusing. The manuscript goes to page 21 and then the tables start on page 1 and the ‘Discussion’ section starts on page 3 or 4. Please correct this.

Response: We apologise that we did not notice this confusing page numbering. This has been corrected. 

7.Reviewer #2: This is a New Zealand-specific study that describes the healthfulness of processed retail foods by food category and manufacturer. The summary statistics are adequately presented and described. Although this study is not a modeling study, the data it presents can be used by researchers to motivate modeling studies that explain, for example, why some consumers chose highly processed foods and why some manufacturers choose to reformulate while others do not.

Response: We acknowledge the reviewer’s comment that this is not a modelling study and that Nutritrack data, alongside national nutrition surveys, could be used in modelling studies. We hope that our study`s findings stimulate future modelling studies, particularly of studies that assess the impact of reducing the availability of highly processed foods on population`s health. A comment had been made in the discussion.

Lines 515-516: The results of this study could stimulate modelling studies such as assessing the impact of reducing the availability of highly processed foods on the health of populations.

8. One important limitation of this study that has not been acknowledged is that the reported product and manufacturer market shares are all based on product counts. A more conventional and informative approach is to weight products by sales volume. This requires retail scanner data that should be available from retailers, if not already from Nielsen. There are likely to be large differences between the weighted and unweighted the summary statistics. For example, Coke Classic is counted equally as Foxton Fizz even though the two have very different sales volume.

Response: We agree with the reviewer that findings from this study could be complemented with sales data which could provide useful information on consumer purchasing patterns in relation to the healthiness of the food supply. However, the cost of these data is prohibitive to many researchers, and consumer purchasing habits were not within the scope of this study. In line with similar previous studies, we report here data on monitoring of the healthiness of the packaged food supply. This is a global recommendation of the Food Monitoring Group (Dunford et al 2012, ref 31). We have indicated in the discussion that future studies including weighed sales data would complement findings from our investigation. We have also added a reference to a NZ study which found that store survey data provided a reasonable estimate of average population exposure to key nutrients from packaged food compared to household panel food purchases.

Line 516-519: Ideally, future assessments of the packaged food supply would be complemented with sales data to provide useful information on consumer purchasing patterns in relation to the healthiness of the food supply. 

Lines 519-522: A comparison of estimates of nutrient exposure from a store survey of packaged foods with those from household panel food purchases in NZ found that store survey data provided a reasonable estimate of average population exposure to key nutrients from packaged foods (44).

Eyles H, Neal B, Jiang Y, Ni Mhurchu C. Estimating population food and nutrient exposure: a comparison of store survey data with household panel food purchases. Br J Nutr. 2016 May 28;115(10):1835–42. Available from: https://www.cambridge.org/core/product/identifier/S000711451600088X/type/journal_article

Reviewer #3: This paper examines which companies dominate the packaged food supply of New Zealand and how healthy their products are using three different metrics. It is a descriptive paper that contains some new and interesting information. My comments are not substantial:

9. Line 67, what kinds of policies are the authors discussing here. Are these labeling, nutrition info disclosing, reformulation etc? It will be desirable if the authors provide some more details here plus some policy background in NZ, which could add a bit to the motivation of the study. As I read more into the end of the paper, I realize the product reformulation is the main policy angle. Still a bit background on New Zealand related food policy development will be helpful (in the context of international comparison as well).

Response: Thank you for this suggestion. We have added information on the NZ policy background in the introduction. Since this manuscript was submitted to Plos One, we have published a report of the Food-EPI assessment of New Zealand government policies and infrastructure support (*). Two of the domains assessed are particularly relevant to this paper: food composition, food labelling. The following paragraph has been added to the introduction.

* Mackay S, Sing F, Gerritsen S., Swinburn B. Benchmarking Food Environments 2020: Progress by the New Zealand Government on implementing recommended food environment policies & priority recommendations. Auckland; 2020.

Line 73-82: A recent 2020 assessment of Government food environment policies and infrastructure support [20] concluded that New Zealand is lagging behind other nations in implementing several major policies to improve food environments. NZ lacks clear government leadership and has no guiding strategy. Although New Zealand has a regulatory system for labelling of health and nutrient claims [21] and the New Zealand and Australian governments have implemented a voluntary front-of-pack labelling system (the Australasian Health Star Rating system based on a nutrient profiling model and implemented in 2014) [22], there are no Government-led targets for nutrients of concern for either packaged food or out-of-home meals. However, the Heart Foundation implemented a voluntary food reformulation programme in 2007 [23], funded by the Ministry of Health.

10. Line 367, I like the international comparison but does the US use the HSR system? A bit clarification is needed here.

Response: The US does not use the Health Star Rating system. The authors (Baldridge et al 2019, ref 32) stated ‘there is no international consensus on the superiority of one particular nutrient profiling model, in part due to the different purposes and contexts for which each model has been developed. One of the most widely used nutrient profile models is that underpinning the Health Star Rating system.’ Clarification has been added to the discussion:

Lines 389-392: Although the Health Star Rating system is only present on foods in Australasia, this system was used in the US study as the system underpinning it is the basis for several similar systems used globally (33), including the UK Food Standards Agency Nutrient Profiling Model (39).

11. Related to my point 1, the policy section seems a bit weak in terms of implications.

Response: We agree and have added the recommended actions of the Food-Epi 2020 assessment to strengthen the implications section. 

Line 483-496: Such solutions include the setting of government-led reformulation targets for processed foods, monitoring population intake targets for nutrients of concern such as sodium and sugar, mandatory front-of-pack labelling, regulation of marketing of unhealthy food, a substantial tax on sugary drinks, and policies around food provision. These are all actions that were prioritised by a public health expert panel that assessed Government progress on food environment policies and prioritised actions in NZ [20]. For reformulation, the Panel recommended a two-tier system for reducing sodium and added sugar in key food categories i.e. (1) setting mandatory maximum levels, similar to the voluntary Heart Foundation targets, to remove the products with the highest levels of sodium and added sugar in product categories, and (2) setting and monitoring targets for voluntary reductions in sales-weighted averages in key food groups as a collective target with the aim of reducing the average sodium or sugar content in a category

12.While revising your submission, please upload your figure files to the Preflight Analysis and Conversion Engine (PACE) digital diagnostic tool, https://pacev2.apexcovantage.com/. PACE helps ensure that figures meet PLOS requirements. To use PACE, you must first register as a user. Registration is free. Then, login and navigate to the UPLOAD tab, where you will find detailed instructions on how to use the tool. If you encounter any issues or have any questions when using PACE, please email PLOS at figures@plos.org. Please note that Supporting Information files do not need this step. 

Response: The figures have been uploaded to PACE

---

## [Decision Letter · Decision Letter 1]

10 Dec 2020

PONE-D-20-17346R1

Which companies dominate the packaged food supply of New Zealand and how healthy are their products?

PLOS ONE

Dear Dr. Mackay,

Thank you for submitting your manuscript to PLOS ONE. At this point, all the reviewers are satisfied with your paper. However, after consulting with the associate editor, I suggest you to address the following comments, which are minor. After your revision, I will not send the paper back for additional review. Please address the comments carefully. 

•       How can other interested researchers gain access to the Nutritrack database used by the Authors? The authors should provide a link, or comment on data are publicly available. Similarly, the minimal dataset should be provided.

•       For reproducibility reasons, the authors should provide a complete list of exclusion criteria for products that were not included in the analysis (specifically the food categories; the other eligibility criteria seem clear ). This can be included in an appendix.

•       Some of the conclusions should be modified to be more precise; for example, the sentence “The packaged food supply in NZ supermarkets is generally unhealthy” should be changed to “The packaged food supply included in the Nutritrack database from NZ supermarkets is generally unhealthy.”

•       There are some specific policy recommendations in the discussion and conclusions that do not seem directly linked or supported by the results, and that should be changed (for example, “For these categories, smaller portion sizes may be more realistic than reformulation and further monitoring should focus on how to describe and capture changes in portion size.” “Efforts to improve the healthiness of products, and promotion of healthier options should be directed at dominant companies because of their high market share”, “Therefore, government leadership is required to make substantive gains across the food supply and address sub-optimal nutrient intake by setting reformulation targets for core food groups, by setting population nutrient intake targets, and by mandating the HSR on products”). Similarly, several of the suggestions found in the Implications section are quite policy focused, and not directly related to the study.

•       The discussion of the BIA-Obesity analysis starting on line 445 does not seem within the scope of this paper – there isn’t a discussion of the methodology used to compare progress against the BIA-Obesity targets nor any timeframe for meeting the targets. I would recommend that this entire section be removed.

We look forward to receiving your revised manuscript.

Kind regards,

Zhifeng Gao

Academic Editor

PLOS ONE

Reviewers' comments:

Reviewer's Responses to Questions

**Comments to the Author**

1. If the authors have adequately addressed your comments raised in a previous round of review and you feel that this manuscript is now acceptable for publication, you may indicate that here to bypass the “Comments to the Author” section, enter your conflict of interest statement in the “Confidential to Editor” section, and submit your "Accept" recommendation.

Reviewer #1: All comments have been addressed

Reviewer #2: All comments have been addressed

Reviewer #3: All comments have been addressed

2. Is the manuscript technically sound, and do the data support the conclusions?

Reviewer #1: (No Response)

Reviewer #2: Yes

Reviewer #3: Yes

3. Has the statistical analysis been performed appropriately and rigorously? 

Reviewer #1: (No Response)

Reviewer #2: N/A

Reviewer #3: Yes

4. Have the authors made all data underlying the findings in their manuscript fully available?

Reviewer #1: (No Response)

Reviewer #2: Yes

Reviewer #3: Yes

5. Is the manuscript presented in an intelligible fashion and written in standard English?

Reviewer #1: (No Response)

Reviewer #2: Yes

Reviewer #3: Yes

6. Review Comments to the Author

Reviewer #1: (No Response)

Reviewer #2: 1. First sentence of the abstract "Improvement of national food supplies are an opportunity to improve a country’s

health." should be "Improvement of national food supplies is...".

2. In abstract, "Products displaying the HSR on the package had a higher mean HSR + SD", please define "+ SD".

Reviewer #3: (No Response)

7. PLOS authors have the option to publish the peer review history of their article (what does this mean?). If published, this will include your full peer review and any attached files.

Reviewer #1: No

Reviewer #2: No

Reviewer #3: **Yes: **Yuqing Zheng

---

## [Author Response · Author response to Decision Letter 1]

17 Dec 2020

Response to Editor

How can other interested researchers gain access to the Nutritrack database used by the Authors? The authors should provide a link, or comment on data are publicly available. Similarly, the minimal dataset should be provided.

The following text has been added to the online submission. This is similar to the explanation provided in a recent paper published in PLOS ONE which also used the Nutritrack database. Bablani et al https://doi.org/10.1371/journal.pmed.1003427

‘Because of the commercial and legal restrictions to the use of copyrighted material it is not possible to share data openly which reveal the product or company names but unredacted versions of the dataset are available with a licensed agreement that they will be restricted to non-commercial use. For access to Nutritrack, please contact the National Institute for Health Innovation at the University of Auckland at enquiries@nihi.auckland.ac.nz’. 

• For reproducibility reasons, the authors should provide a complete list of exclusion criteria for products that were not included in the analysis (specifically the food categories; the other eligibility criteria seem clear ). This can be included in an appendix.

A supplementary table has been added (S1 Table) which provides the specific food categories excluded. Consequently the original S1 Table has been renamed S2 Table. 

• Some of the conclusions should be modified to be more precise; for example, the sentence “The packaged food supply in NZ supermarkets is generally unhealthy” should be changed to “The packaged food supply included in the Nutritrack database from NZ supermarkets is generally unhealthy.”

The following sentence has been altered in the abstract and the conclusion to specify that this is about foods included in the Nutritrack database..

Line 34: The New Zealand supermarket packaged food supply included in the Nutritrack database is dominated by a small number of companies and is mostly unhealthy

Line 547-548: In 2018, packaged foods and beverages available in NZ supermarkets included in the Nutritrack database were dominated

• There are some specific policy recommendations in the discussion and conclusions that do not seem directly linked or supported by the results, and that should be changed (for example, “For these categories, smaller portion sizes may be more realistic than reformulation and further monitoring should focus on how to describe and capture changes in portion size.” “Efforts to improve the healthiness of products, and promotion of healthier options should be directed at dominant companies because of their high market share”, “Therefore, government leadership is required to make substantive gains across the food supply and address sub-optimal nutrient intake by setting reformulation targets for core food groups, by setting population nutrient intake targets, and by mandating the HSR on products”). Similarly, several of the suggestions found in the Implications section are quite policy focused, and not directly related to the study.

“For these categories, smaller portion sizes may be more realistic than reformulation and further monitoring should focus on how to describe and capture changes in portion size.” 

Line 430: This recommendation and the sentence proceeding it has been deleted as portion size was not a focus of this analysis. 

“Efforts to improve the healthiness of products, and promotion of healthier options should be directed at dominant companies because of their high market share”,

We have considered the comment about the recommendations and if possible, we would prefer to retain this recommendation as we think it is relevant to the paper which is about the dominant companies. The Heart Foundation in New Zealand is the main organisation that works with companies to encourage reformulation, and deliberately works with the companies and products with higher market share as this will have the most impact on population health.

“Therefore, government leadership is required to make substantive gains across the food supply and address sub-optimal nutrient intake by setting reformulation targets for core food groups, by setting population nutrient intake targets, and by mandating the HSR on products”

Line 643: We have removed ‘population nutrient intake targets’ as this is less directly linked to the paper. The remainder of the recommendation is retained as I do believe that this recommendation is relevant as the food supply in New Zealand means there are a lot of unhealthy foods available which demonstrates there is scope for reformulation targets, and the HSR is displayed selectively on healthier products so mandating the HSR would address this. 

Line 492: In the implications section, I have removed some of the policy solutions “regulation of marketing of unhealthy food” and “policies around food provision”.

• The discussion of the BIA-Obesity analysis starting on line 445 does not seem within the scope of this paper – there isn’t a discussion of the methodology used to compare progress against the BIA-Obesity targets nor any timeframe for meeting the targets. I would recommend that this entire section be removed.

This section has been removed.

---

## [Editor Report · Decision Letter 2]

28 Dec 2020

Which companies dominate the packaged food supply of New Zealand and how healthy are their products?

PONE-D-20-17346R2

Dear Dr. Mackay,

We’re pleased to inform you that your manuscript has been judged scientifically suitable for publication and will be formally accepted for publication once it meets all outstanding technical requirements.

Kind regards,

Zhifeng Gao

Academic Editor

PLOS ONE
---

## [Editor Report · Acceptance letter]

15 Jan 2021

PONE-D-20-17346R2 

Which companies dominate the packaged food supply of New Zealand and how healthy are their products? 

Dear Dr. Mackay:

I'm pleased to inform you that your manuscript has been deemed suitable for publication in PLOS ONE. Congratulations! Your manuscript is now with our production department. 

Kind regards, 

on behalf of

Dr. Zhifeng Gao 

Academic Editor

PLOS ONE